

# Increasing evidence that bats actively forage at wind turbines

Cecily F. Foo[1], Victoria J. Bennett[2], Amanda M. Hale[1], Jennifer M. Korstian[1], Alison J. Schildt[1] and Dean A. Williams[1]

[1] Department of Biology, Texas Christian University, Fort Worth, TX, United States of America
[2] School of Geology, Energy & the Environment, Texas Christian University, Fort Worth, TX, United States of America

Corresponding author
Amanda M. Hale, a.hale@tcu.edu

## ABSTRACT

Although the ultimate causes of high bat fatalities at wind farms are not well understood, several lines of evidence suggest that bats are attracted to wind turbines. One hypothesis is that bats would be attracted to turbines as a foraging resource if the insects that bats prey upon are commonly present on and around the turbine towers. To investigate the role that foraging activity may play in bat fatalities, we conducted a series of surveys at a wind farm in the southern Great Plains of the US from 2011–2016. From acoustic monitoring we recorded foraging activity, including feeding buzzes indicative of prey capture, in the immediate vicinity of turbine towers from all six bat species known to be present at this site. From insect surveys we found Lepidoptera, Coleoptera, and Orthoptera in consistently high proportions over several years suggesting that food resources for bats were consistently available at wind turbines. We used DNA barcoding techniques to assess bat diet composition of (1) stomach contents from 47 eastern red bat (*Lasiurus borealis*) and 24 hoary bat (*Lasiurus cinereus*) carcasses collected in fatality searches, and (2) fecal pellets from 23 eastern red bats that were found on turbine towers, transformers, and tower doors. We found that the majority of the eastern red bat and hoary bat stomachs, the two bat species most commonly found in fatality searches at this site, were full or partially full, indicating that the bats were likely killed while foraging. Although Lepidoptera and Orthoptera dominated the diets of these two bat species, both consumed a range of prey items with individual bats having from one to six insect species in their stomachs at the time of death. The prey items identified from eastern red bat fecal pellets showed similar results. A comparison of the turbine insect community to the diet analysis results revealed that the most abundant insects at wind turbines, including terrestrial insects such as crickets and several important crop pests, were also commonly eaten by eastern red and hoary bats. Collectively, these findings suggest that bats are actively foraging around wind turbines and that measures to minimize bat fatalities should be broadly implemented at wind facilities.

# INTRODUCTION

Unlike conventional sources of energy such as oil, gas, and coal, utility-scale wind farms require no fuel, do not consume water, and produce no greenhouse gas emissions or

other pollutants during the energy production phase. In 2013, wind power supplied 4.5% of the electrical energy consumed in the United States (US) and the US Department of Energy's goal is to increase this percentage to at least 20% by 2030, providing substantial environmental and economic benefits from a sustainable, domestic energy source (*USDOE, 2015*). Despite these recognized benefits, wind energy development has drawbacks; for example, annual wind-related bat fatality is estimated in the hundreds of thousands of bats (*Cryan, 2011*; *Arnett & Baerwald, 2013*; *Smallwood, 2013*) with projected increases as wind energy development continues (e.g., *Zimmerling & Francis, 2016*). Consequently, wildlife conservation has become an important consideration in the expansion of wind power.

Migratory tree bats, particularly lasiurine species, have the highest mortality rates at wind facilities in North America, which peak from midsummer to early fall and coincide with the bats' seasonal migration (*Kunz et al., 2007*; *Arnett et al., 2008*; *Arnett & Baerwald, 2013*; *Hein & Schirmacher, 2016*). More than 75% of wind-related bat fatalities are comprised of three species: hoary (*Lasiurus cinereus*), eastern red (*Lasiurus borealis*), and silver-haired (*Lasionycteris noctivagans*) bats. We know little about the migratory behavior or the population status of these tree bats, but there is increasing concern that high fatality rates at wind turbines could have long-term effects on bat populations (*Kunz et al., 2007*; *Arnett et al., 2008*; *Cryan & Barclay, 2009*; *Arnett & Baerwald, 2013*; *Jameson & Willis, 2014*; *Frick et al., 2017*).

Although the proximate causes of bat fatality at wind turbines are relatively well understood (i.e., bats may die from barotrauma (*Baerwald et al., 2008*; *Grodsky et al., 2011*; but see *Rollins et al., 2012*), collision with the rotating blades (*Horn, Arnett & Kunz, 2008*), or a combination of the two), the ultimate causes are still unclear (*Kunz et al., 2007*; *Arnett et al., 2008*; *Cryan & Barclay, 2009*). Nevertheless several lines of evidence suggest that bats may be attracted to wind turbines and a number of specific hypotheses have been proposed to explain this phenomenon. One possibility is that bats find something about the turbines themselves to be interesting (*Cryan & Barclay, 2009*). For example, red aviation lights on top of turbine towers have been considered to be a potential source of interest to bats; however, studies have shown that mortality at towers with aviation lights is similar to or even less than mortality at towers without aviation lights (*Arnett et al., 2008*; *Baerwald, 2008*; *Bennett & Hale, 2014*). Alternatively, bats may misperceive wind turbines to be a resource. For example, a study by *Cryan et al. (2014)* suggested that tree bats, in particular, may misperceive turbines to be trees and are therefore attracted to the turbines to seek roosting and mating opportunities. Another study hypothesized that bats may misperceive wind turbine towers as water (*McAlexander, 2013*), as previous research has shown that echolocating bats misidentify artificial smooth surfaces to be water (*Greif & Siemers, 2010*; *Russo, Cistrone & Jones, 2012*). Another possible explanation is that wind turbines may actually provide bats with resources, such as water (as condensation on the tower), roosting, and foraging opportunities. To date, no published study has demonstrated that wind turbines provide bats with water, but a recent study by *Bennett, Hale & Williams (2017)* shows that bats will roost on turbines and evidence from recent studies based on stomach content analyses (e.g., *Valdez & Cryan, 2013*; *Rydell et al., 2016*), acoustic monitoring (e.g., *McAlexander, 2013*), and thermal imagery

(e.g., *Horn, Arnett & Kunz, 2008*; *McAlexander, 2013*; *Cryan et al., 2014*), indicate that bats are actively foraging near wind turbines. It is also important to recognize that these various attractors are not mutually exclusive and their relative importance likely varies by species, time of year, and geographic location.

Despite the emerging evidence in support of the foraging attraction hypothesis, we still do not know the extent to which bats are feeding on insects that are present on and around wind turbine towers. To address this need, we conducted a multifaceted study to investigate the foraging attraction hypothesis at a wind facility in the southern Great Plains, US. This region has some of the highest installed wind capacity in the continental US (*Wiser & Bolinger, 2016*), and yet remains relatively understudied with respect to a wide range of wind-wildlife issues. *Long, Flint & Lepper (2011)* found that insects are drawn to light-colored turbines in particular; and as turbines are commonly painted with light colors, bats may be attracted to wind farms as a result of insect aggregations on and around the turbine towers. Herein, we examine multiple lines of evidence with respect to bats foraging at wind turbines, specifically addressing several predictions of the "attraction to insect aggregations" hypothesis as outlined by *Cryan & Barclay (2009)*. First, we used acoustic monitoring to determine if bats were successfully foraging near turbine towers, as evidenced by feeding buzzes (listed as the first prediction in *Cryan & Barclay, 2009*). Second, we used insect surveys to ascertain if there were foraging opportunities for bats near turbine towers (listed as the third prediction in *Cryan & Barclay, 2009*). Third, we utilized bat carcasses collected during fatality searches to determine if the bats had full stomachs at the time of death (listed as the fourth prediction in *Cryan & Barclay, 2009*) and to identify (using DNA barcoding) what the bats had been eating prior to being killed. Fourth, we utilized bat fecal pellets collected during turbine searches as a second source of material to identify (using DNA barcoding) what the bats had been eating. And fifth, we compared insects found in the bats' stomachs and fecal pellets to the insects found at the turbines (listed as the fifth prediction, *Cryan & Barclay, 2009*). Recent studies have shown that when aerial-hawking bats, such as the eastern red and hoary, have full stomachs, they do not fly far from their foraging sites to find a suitable roost to digest their food (*Knight & Jones, 2009*; *Lison, Palazon & Calvo, 2013*; *Montero & Gillam, 2015*). Thus, if the stomachs of bat carcasses were full and the prey species were also found near turbine towers, this would suggest that the bats were likely foraging in the vicinity of the wind turbines prior to death.

## MATERIALS & METHODS

### Study site

We conducted our study at Wolf Ridge Wind, LLC (hereafter Wolf Ridge), a utility-scale wind farm owned and operated by NextEra Energy Resources in the southern Great Plains of the US (N33°44′01.19″, W97°24′57.26″). The 48-km$^2$ wind resource area comprises 75 1.5-MW General Electric wind turbines (GE 1.5xle specifications: 80-m tower, three 42-m blades attached to the front of the nacelle, 84-m diameter rotor swept zone that reaches 122-m above ground) situated in a matrix of cattle-grazed pastures, hayfields, and some

agricultural lands, with shrub-woodland habitat extending from the riverine valleys of the Red River escarpment to the north. Wolf Ridge has been operational since October 2008, and based on fatality monitoring surveys and acoustic surveys conducted from 2009 to 2014, six bat species are known to be present at this site: eastern red bat, hoary bat, silver-haired bat, tri-colored bat (*Perimyotis subflavus*), evening bat (*Nycticeius humeralis*), and Mexican free-tailed bat (*Tadarida brasiliensis*) (*Bennett & Hale, 2014*).

## Acoustic monitoring

We conducted acoustic surveys at wind turbines to determine which bat species were in close proximity to wind turbine tower surfaces and to characterize bat activity at wind turbines. Over a seven year period in conjunction with other surveys (e.g., fatality searches or thermal camera studies), we recorded bat activity using two different acoustic monitoring systems in close proximity to wind turbine tower surfaces. Between 28 July 2010 and 22 July 2011, we deployed two ReBAT$^{TM}$ acoustic monitoring systems ∼85 m above ground on the rear of the nacelle of two operational wind turbines. Each ReBAT$^{TM}$ system comprised two acoustic detectors, one facing upward from the nacelle into the upper half of the rotor swept zone (RSZ) and the other facing downward into the lower half of the RSZ. From 14 May to 1 October 2012, 5 April to 24 October 2013, 9 July to 28 September 2015, and 1 July to 10 August 2016, we used Binary Acoustic Technology (BAT, Tucson, AZ) AR125-EXT ultrasonic receivers and BAT FR125 recorders mounted to tripods at the base of the wind turbines to record bat activity. We positioned this latter acoustic detector assembly on the gravel pad approximately 2 m from the turbine base with the AR125 receiver pointing 45° toward the tower surface to record bat activity within close proximity of the wind turbine tower surface.

Both types of acoustic detector assemblies recorded sound files as a standard .wav file. Furthermore, from 2010 to 2012 detectors were set up to begin recording from dusk until dawn, whereas in 2013, 2015, and 2016 detectors recorded data starting at dusk and continued recording for three hours, a time interval considered to be a primary foraging activity period of bats (*Baerwald & Barclay, 2011*).

We used SonoBat bat call analysis software (version 3.04) to analyze the recorded sound files. Each file containing a bat call was counted as one bat pass (*Miller, 2001*). Using full-spectrum spectrograms generated in Sonobat we manually determined each call to species and activity (where possible) using available call libraries. For the latter, we identified the following four distinct activities: *commuting*—consecutive calls (i.e., individual chirps) were synchronized with wing beats (*Altringham, 2011*) and were either constant, steadily decreasing, or steadily increasing in call strength (in addition any sound file with <2 calls was also categorized as commuting; e.g., a bat moving through the area from one foraging site to another); *searching*—consecutive calls were synchronized with wing beats, but varied in strength due to the bat turning its head from side to side while echolocating (*Altringham, 2011*); *foraging or approach phase*—call interval varied with multiple calls occurring in succession within a single wing beat, and call strength was constant, steadily decreasing, or steadily increasing (*Altringham, 2011*); and *feeding or terminal buzz*—interval between successive calls decreased rapidly and the frequency of these calls was higher or lower

(depending on species) than calls representing the other three activities (*Altringham, 2011*). The latter two acoustic activities are generally associated with foraging behavior, as the *foraging or approach phase* of echolocation calls is an indication that bats are in pursuit of prey, and *feeding or terminal buzzes* suggest that bats are successfully capturing prey.

## Insect surveys

We used light traps and malaise traps to survey insects at three pairs of turbines at Wolf Ridge in 2012 (*Cochran, 2013*), 2013 and 2015). Light traps use a UV light that attracts nearby insects and typically captures a large variety of species, whereas malaise traps have no attractant and typically catch a more limited sample with fewer individuals and fewer species represented (*Hosking, 1979*). Light trapping therefore provided us with a general characterization of the insect community at the wind turbines, whereas malaise trapping functioned as a passive control that would inform us if we were missing species with the light traps that were otherwise present at the study site.

We conducted insect surveys at two turbines a night, three nights a week, over a six week period in July and August during each year. We selected this time period as it coincides with peak bat fatality at our site (*Bennett & Hale, 2014*). We were able to light trap as long as wind speeds were <15 mph and there was no precipitation. We were able to employ malaise traps as long as wind speeds were <10 mph and there was no precipitation.

In 2012, surveys took place in two 3-hour periods, the first beginning at dusk and the second beginning 3 h before dawn, as these times have been shown to coincide with peak bat foraging activity (*Baerwald & Barclay, 2011*). In 2013, surveys began at dusk and ran continuously through the night until dawn. While in 2015, we streamlined the survey method and only sampled insects for a 3-hour period beginning at dusk.

We assembled our light traps on the gravel pad surrounding the turbine tower. Light traps consisted of a fluorescent black bulb shielded by opaque plastic on three sides: the side facing the turbine and the top were not shielded, illuminating the turbine tower. The light trap was placed on a white sheet on the gravel pad, which allowed us to see and collect insects more easily. We assembled malaise traps on the ground next to the gravel pad (≤5 m from the turbine) on the opposite site of the turbine from the light trap, where they would be shielded from the UV light. Because we left the traps out for different lengths of time and checked them at different time intervals each year, we cannot directly compare insect abundance and diversity among years. Instead, we use the results of the insect surveys to characterize the insect community at wind turbines at the time of year when bat fatality rates are the highest.

We collected and tallied each insect during the survey period by morpho-species in the field. In order to avoid counting individual insects multiple times, we collected the insects as we counted them and released them at the end of the survey period each night. For each survey night, unique specimens were photographed, identified to order, and preserved in either glycine bags or 100% ethanol.

## Bat carcasses and assessment of bat stomach fullness

Although six bat species have been found in fatality monitoring surveys, we only included eastern red bat and hoary bat carcasses in this study because these species had the highest

fatality rates at Wolf Ridge (*Bennett & Hale, 2014*), and experience high levels of fatality at wind facilities across North America (e.g., *Arnett & Baerwald, 2013*; *Zimmerling & Francis, 2016*). To select carcasses for analysis, we prioritized those in best overall condition (i.e., no obvious decay or damage by insect scavengers), including adult and juvenile males and females of both species that were collected between July and August of 2013 and 2014 and subsequently stored at −4 °C. Thus, we dissected 45 eastern red bat (27 females, 18 males; 40 adults, five juveniles) and 23 hoary bat carcasses (11 females, 12 males; 19 adults, four juveniles), removing their digestive systems (esophagus, stomach, and intestines) and storing them with their contents intact in 70% ethanol.

Before beginning the genetic analyses, we separated the bat stomachs from the esophagi and intestines and visually determined if the stomachs were full using the following definitions. A stomach was considered "full," if it appeared taut from the outside (i.e., not folded or wrinkled), whereas a stomach with obvious extra space or folded membrane was considered "not full". The stomach contents from each bat were then homogenized using a mortar and pestle and weighed. In some instances the stomach membrane was perforated and the contents had been exposed to ethanol, so we allowed the ethanol to evaporate for up to an hour in the extraction hood (described below) prior to homogenization and weighing. We conducted a two-sample $t$-test to determine if stomachs classified as "full" were significantly heavier than "not full" stomachs.

## Genetic analysis of bat stomach contents

We extracted DNA from the homogenized stomach contents using DNeasy® mericon Food Kits (Qiagen, Valencia, CA, USA; *Zarzoso-Lacoste, Corse & Vidal, 2013*). We included a negative control with each round of extraction (3 to 7 bat stomach samples) to ensure non-contamination of reagents. All extractions were completed in a dedicated extraction AirClean® 600 PCR workstation to minimize contamination.

Samples were then amplified using arthropod-specific primers ZBJ-ArtF1c and ZBJ-ArtR2c developed by *Zeale et al. (2011)*. We set up the polymerase chain reactions (PCR) in a dedicated PCR AirClean® 600 PCR workstation in a different room from where the DNA extractions took place. Again, we included negative controls in our PCR reaction batches. PCR (10 μL) contained 2 μl DNA, 0.5 μM of each primer, 1X Qiagen Multiplex PCR Master Mix with HotStarTaq, Multiplex PCR buffer with 3 mM $MgCl_2$ pH 8.7, and dNTPs. Reactions were cycled in an ABI 2720 thermal cycler. PCR ran for one cycle at 95 °C for 15 min, followed by 40 cycles of 30 s at 94 °C, 90 s at 55 °C, 90 s at 72 °C, and then a final extension at 72 °C for 5 min. We purified the products on a gel, ligated them into pGEM-T vectors (Promega, Madison, WI, USA), and then transformed them into JM109 competent cells. We plated the transformed cells on ampicillin plates and left them in a 37 °C incubator overnight. The following day we selected colonies that had been successfully transformed (i.e., those in which the PCR product had been inserted) based on color (white colonies had been transformed, whereas blue colonies had not). Each clone was amplified using vector-specific primers (F: CGACTCACTATAGGGCGAATTG, R: CTCAAGCTATGCATCCAAGG). Unincorporated nucleotides and excess primers were removed from PCR products using *ExoI* and *rSAP* (New England Biolabs, Ipswich,

MA, USA) according to manufacturer protocols. PCR products were then unidirectionally sequenced using the forward vector primers and ABI Big Dye Terminator Cycle Sequencing v3.1 Chemistry (Life Technologies, Carlsbad, CA, USA). We electrophoresed sequences on an ABI 3130XL Genetic Analyzer (Life Technologies, Carlsbad, CA, USA), edited and trimmed the sequences using Sequencher v5.0 (Gene Codes Ann Arbor, MI, USA), and then aligned the sequences in MEGA 6.0 using Muscle (*Edgar, 2004*; *Tamura et al., 2013*). We used the Barcode of Life Data System (BOLD), an online index of known DNA sequences, to identify sequences (*Ratnasingham & Hebert, 2007*; http://www.boldsystems.org). We assigned species for a 99–100% matched sequence, we assigned genus for a 95–98% match, assigned family for a 90–94% match, and assigned order for an 85–90% match according to the methods in *Clare et al. (2009)* and *Zeale et al. (2011)*. From each stomach sample, we picked and sequenced at least 12 colonies containing recombinant clones since a preliminary study indicated that 10 clones was sufficient to detect all insect species present. In two stomachs, we were only able to sequence nine clones and in one stomach we were only able to sequence eight clones because fewer than 10 recombinant clones were present after cloning or some clones gave low quality sequence.

We created neighbor-joining trees using the Kimura Two-Parameter distance in MEGA to determine the number of OTUs (operational taxonomic units) that were present in the samples for which there was less than a 99% match in BOLD. We created separate trees for each order and classified samples as belonging to different species if they were >2% different and clearly clustered separately from other known species identified in BOLD. We used letters to distinguish unidentified species from one another (e.g., Lepidoptera A and Lepidoptera B).

## Fecal collection, DNA extraction, amplification, and sequencing

From July to November 2011 and April to October 2012, we conducted weekly systematic searches for bat feces at all 75 wind turbines at Wolf Ridge (*Bennett, Hale & Williams, 2017*). Single fecal samples were collected from between the upper slats of the turbine door, between the gills of the transformer, on the frame beneath the gills of the transformer, and beneath the stairwell. Once found, we placed each fecal pellet in a 1.5 ml plastic tube and stored them at room temperature.

DNA extraction followed the protocol outlined in the QIAamp DNA Stool Mini-kit (Qiagen Genomics, Valencia, CA, USA). A negative control (i.e., a tube with no fecal sample in it) was included with each round of extraction to ensure that the extraction reagents used were not contaminated. Two successive PCR procedures were then undertaken. We first used PCR to identify fecal samples to bat species following the methods described in *Korstian et al. (2015)*. As most of the fecal pellets (59% of 56 pellets) were from eastern red bats (<5 were from hoary bats; *Bennett, Hale & Williams, 2017*), we limited diet analysis from fecal pellets to eastern red bats only. Following bat species identification, we then used a second PCR to amplify the remnants of prey items in the fecal pellets from eastern red bats. PCR, cloning, sequencing, and arthropod identification protocols were identical to the ones used for bat stomach contents. For each fecal sample, we picked and sequenced at least 12 colonies containing recombinant clones since a preliminary study indicated that

10 clones was sufficient to detect all insect species present. For two fecal pellets, we were only able to sequence seven and nine clones because less than 10 recombinant clones were present after cloning or some clones gave low quality sequence.

For the clones generated from any given bat stomach or fecal sample, each insect species detected was counted only one time. We calculated the Simpson's diversity index to summarize the number and abundance of prey items separately for eastern red bat stomachs, hoary bat stomachs, and eastern red bat fecal pellets. We present sampling curves for both the stomach and fecal samples using the number of clones successfully sequenced for all samples and the number of insects identified.

### Comparing stomach and fecal contents to insect surveys

To determine whether insect species consistently found in bat stomachs and fecal pellets were also present near turbine towers, we compared all of the insect species identified in BOLD that were found in ≥5 stomach samples to the insect specimens that were collected during surveys in July and August of 2013 and 2015 and identified these species using the voucher specimens we collected. In all cases, the most common species found in bat stomachs were easy to identify morphologically from the insects we collected. If we found that bats were consistently eating insects that were not captured at turbines, it would suggest that bats were foraging elsewhere and not using the turbines as a foraging resource. Conversely, if the species that we consistently found in bat stomachs were present in insect surveys conducted at turbines, then this would provide support for the foraging attraction hypothesis.

## RESULTS

### Acoustic bat activity at wind turbines

Acoustic data were collected at wind turbine towers on 284 nights (93 in 2010, 42 in 2012, 90 in 2013, 38 in 2015, and 21 in 2016). We recorded a total of 3,606 bat passes and identified calls from all six bat species known to be in the study area (Fig. 1). Within this dataset, foraging and approach phase activity were recorded in 23% of the bat passes ($n = 836$) and feeding or terminal buzzes were recorded in 3.1% of the bat passes ($n = 113$). Of these feeding or terminal buzzes, 56% were recorded at detectors placed at the turbine nacelles. All six bat species exhibited foraging behavior at wind turbines (Fig. 1).

### Insect surveys

We confirmed that the majority of the insect orders caught in the malaise traps did not differ from the insect orders collected with light trapping (Fig. S1, Fig. 2), therefore we only summarized the light trapping results to characterize the insect community near the turbine towers. In 2012, we light trapped a total of 17 nights between July and August, collecting 1,238 invertebrates belonging to nine orders. The three most abundant orders were Coleoptera (37.2%), Orthoptera (23.7%), and Lepidoptera (20.0%). In 2013, we light trapped a total of 13 nights between July and August, collecting 1,937 invertebrates belonging to 11 orders. The three most abundant orders were Lepidoptera (42.8%), Coleoptera (38.0%), and Hemiptera (9.1%). In 2015, we light trapped for 16 nights

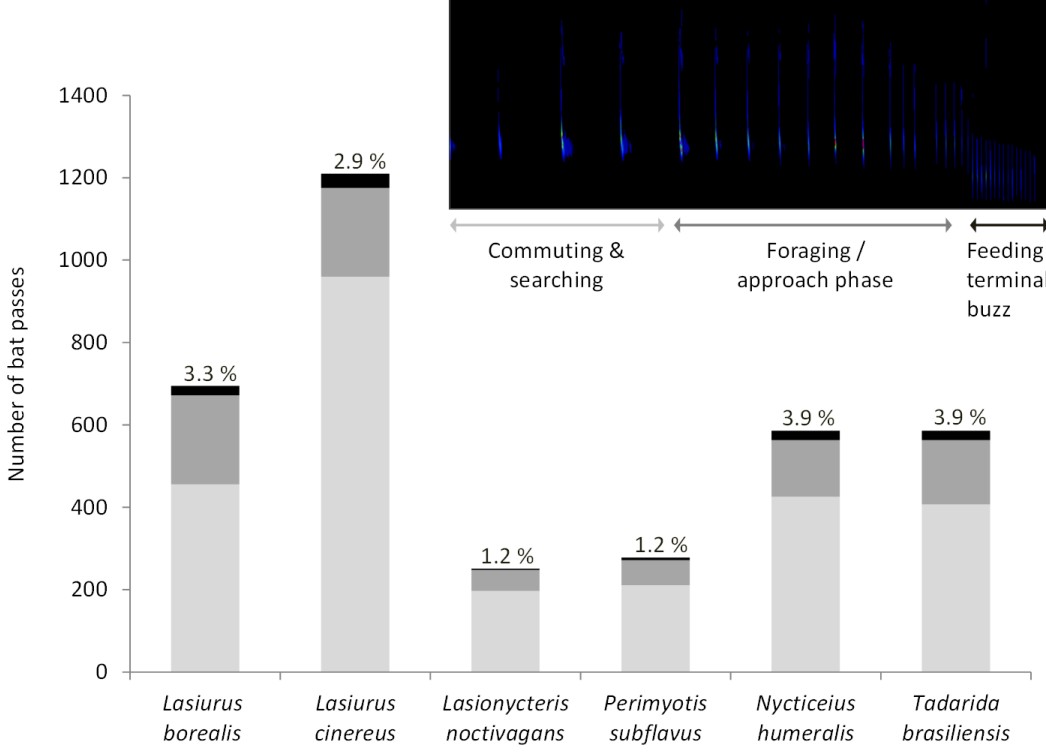

**Figure 1 Number of bat passes recorded at turbine towers.** Number of bat passes, separated into 3 activity categories (light grey, commuting & searching; dark grey, foraging/approach phase; black, feeding/terminal buzz), by species recorded in acoustic surveys at wind turbine towers in 2010, 2012, 2013, 2015, and 2016 at the Wolf Ridge wind farm. The percentage of bat passes that included feeding or terminal buzz activity is included above each species' bar.

between July and August, collecting 7,479 invertebrates belonging to 13 orders. The three most abundant orders were Coleoptera (54.9%), Lepidoptera (14.7%), and Hemiptera (13.2%).

Due to differences in survey methods among years, we could not statistically compare the three years of insect surveys to determine if the insect community at Wolf Ridge changed over time. However, an informal comparison based on the average biweekly proportions of each order suggests that the insect community has remained relatively consistent between July and August of 2012, 2013, and 2015 (Fig. 2). Note that the confidence intervals are wide due to nightly variation in insect abundance among survey periods. For example, on one night we might have not caught any water beetles, whereas the next survey night might have coincided with an emergence of water beetles. Additionally, some species of insects were only captured on a single survey night during the season.

## Assessment of bat stomach fullness

Of the 45 eastern red bats included in this study, 22 had full stomachs (three stomachs were not weighed). The mean $\pm$ SD weight of full stomach contents was 0.139 $\pm$ 0.076 g, whereas not full stomach contents weighed 0.043 $\pm$ 0.039 g ($t = 5.04$, $df = 27$, $P < 0.001$,
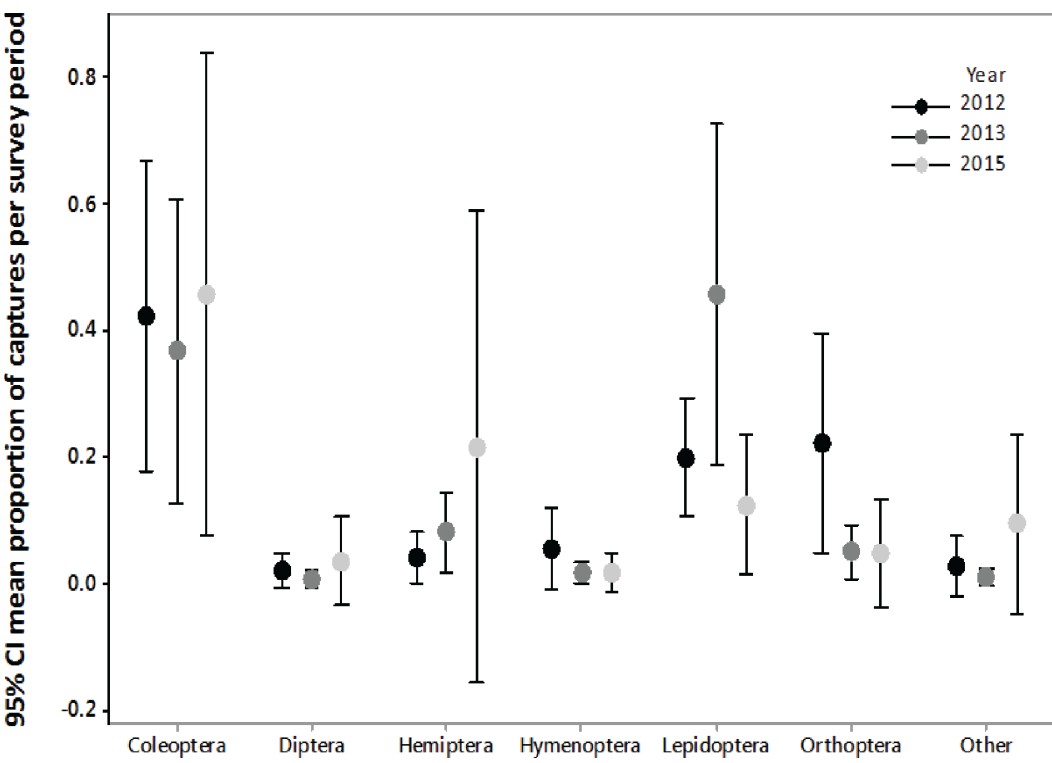

**Figure 2** **Biweekly proportions of insect orders surveyed at turbine towers.** Biweekly averages and 95% CI of the proportions of each order collected during July–August light trapping in 2012, 2013, and 2015 at the Wolf Ridge wind farm. The "other" category includes Blattodea, Ephemeroptera, Neuroptera, Odonata, Plecoptera, Trichoptera, and spiders.

$n = 20$ and 22 stomachs, respectively). Of the 23 hoary bats included in this study, 16 had full stomachs (1 stomach was not weighed). The mean ± SD weight of full stomach contents was 0.458 ± 0.252 g, whereas not full stomach contents weighed 0.131 ± 0.114 g ($t = 4.19$, $df = 19$, $P < 0.001$, $n = 15$ and seven stomachs, respectively). For both eastern red bats and hoary bats, the body masses of juveniles and adults were not significantly different (see Data S1), so we did not separate our analysis by age group.

## Genetic analysis of bat stomach contents

Insect DNA was successfully extracted and amplified from all of the 68 bat stomachs included in this study. The average number of clones for the 68 bat stomachs was 13.4 (range: 8–21 clones). The sampling curve peaked at 10 clones indicating that 10 clones was sufficient to detect all prey species within a single stomach sample (Fig. S2A).

Collectively, the results of our stomach analysis yielded 153 insects representing 60 genetically distinct species. Based on the percentage match to known sequences in BOLD, 38 insects were identified to species, 10 were identified to genus, three were identified to family, and nine were identified to order (Tables S1 and S2). Individual bats in our study had a mean (±SE) of 2.26 ± 0.11 prey species in their stomachs (range: 1–6 species; Fig. 3).

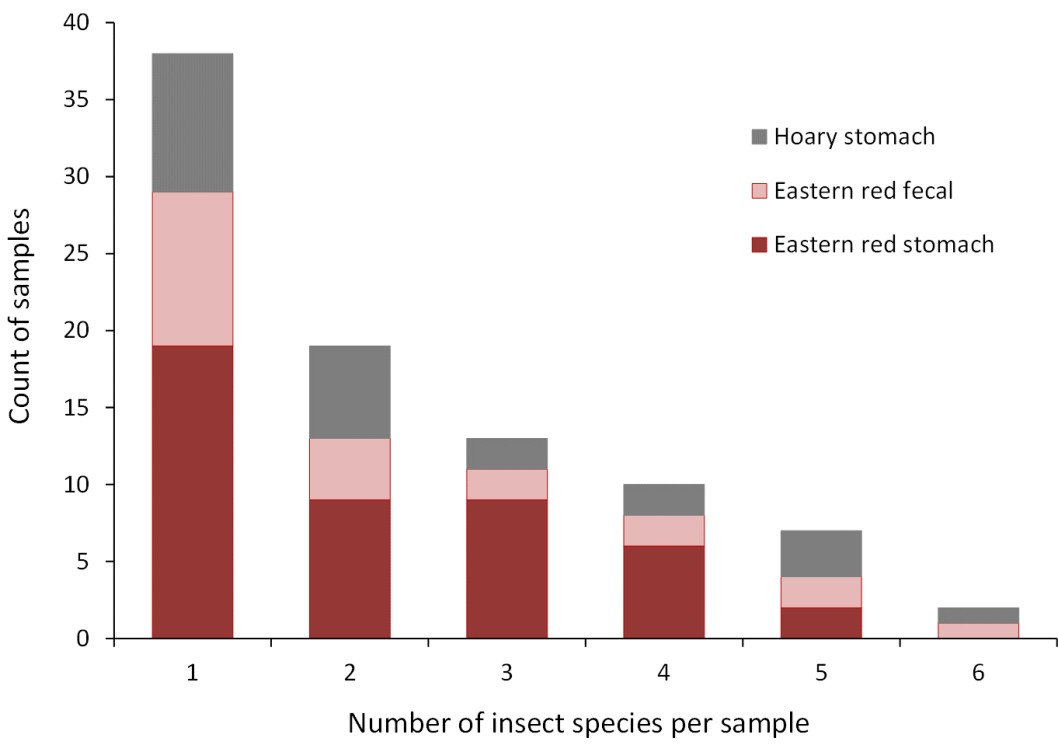

**Figure 3** **Number of insect species found in bat stomachs and fecal pellets.** Number of bat stomach samples ($n = 45$ eastern red bats, $n = 23$ hoary bats) and bat fecal samples ($n = 23$ eastern red bats) in which from one to six different insect species were found.

The majority of our samples consisted of only one or two species of insects (28 bats and 15 bats, respectively).

Eastern red bats ($n = 45$) had a mean (±SE) of 2.2 ± 0.19 individual prey species in their stomachs (range: 1–5 species; Fig. 3). We found 43 different species of insects from seven orders in eastern red bat stomachs (Fig. 4A; Table S1). Thirty-one of these species were detected in only one stomach (Fig. S3). We detected one species of moth (*Spodoptera frugiperda*) in 11 different stomachs and one species of cricket (*Gryllus spp.*) in 29 stomachs (Table S1). Lepidoptera comprised 55.1% and Orthoptera comprised 32.7% of the insect species identified in eastern red bat stomachs. The remainder belonged to Blattodea, Coleoptera, Diptera, Hemiptera, and Neuroptera. In addition to the species mentioned above, the following insects were detected in bats from both years: *Parcoblatta spp.*, *Achyra rantalis*, *Euchromius ocelleus*, and *Bleptina caradrinalis*.

Hoary bats ($n = 23$) had a mean (±SE) of 2.4 ± 0.34 individual prey species in their stomachs (range: 1–6 species; Fig. 3). We found 25 different species of insects from three orders in hoary bat stomachs (Fig. 4B; Table S2). Eighteen of these species were found in only one stomach (Fig. S3). Similar to our eastern red bat stomach analysis, we found that Lepidoptera was the most abundant and diverse order in the stomachs; the most frequently detected moth, *S. frugiperda*, was found in seven stomachs. *Gryllus spp.* were the most frequently detected species and were found in 18 stomachs. Lepidoptera comprised 60.7%

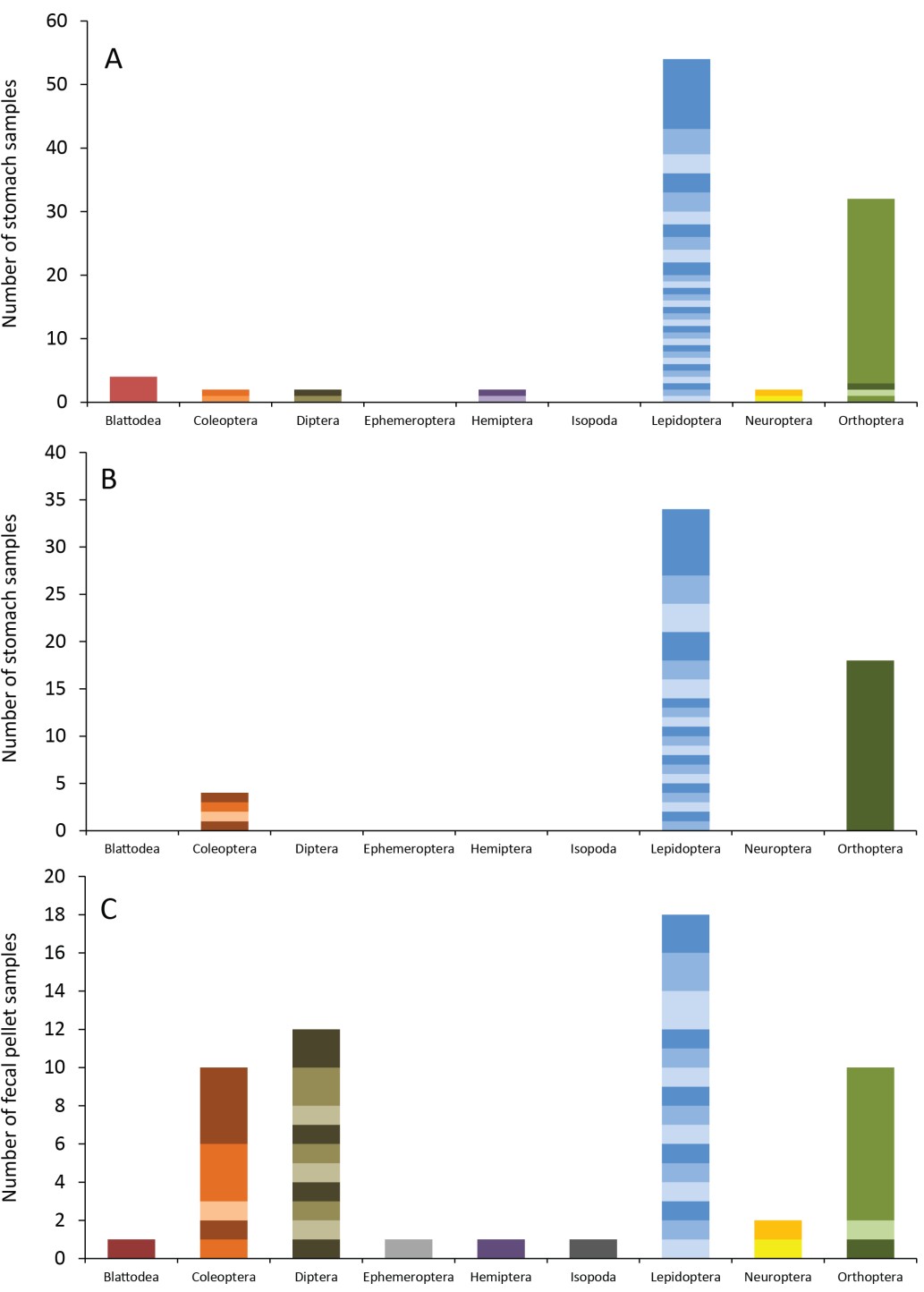

**Figure 4  Insect orders (and species within each) found in bat stomachs and fecal pellets.** Insect orders found in (A) eastern red bat stomach contents ($n = 45$ bats collected in 2013 and 2014), (B) hoary bat stomach contents ($n = 23$ bats collected in 2013 and 2014), and (C) eastern red bat fecal samples ($n = 23$ pellets collected in 2011–2012) from the Wolf Ridge wind farm. Each band in the bar represents a different insect species. Note that the $y$-axis differs in each of the figure (A–C).

and Orthoptera comprised 32.1% of the species identified in hoary bat stomachs. The remainder consisted of four species of Coleoptera, each only detected once. In addition to the species mentioned above, only two other species of moths were detected in both years: *E. ocelleus* and *Helicoverpa zea*.

### Genetic analysis of bat fecal pellets

Insect DNA was successfully extracted and amplified from 23 of the 33 eastern red bat fecal pellets collected from wind turbines (*Bennett, Hale & Williams, 2017*). The average number of clones for the fecal pellets was 13.5 (range: 7–18 clones), and the sampling curve appeared to level off at approximately 10 clones; however, a few samples were highly diverse causing the curve to continue to increase after 14 clones (Fig. S2B).

Collectively, the results of our fecal analysis yielded 57 insects representing 39 genetically distinct species. Based on the percentage match to known sequences in BOLD, 18 insects were identified to species, 10 were identified to genus, five were identified to family, and six were identified to order (Table S3). Individual fecal pellets in our study contained a mean ( $\pm$ SE) of 2.4 $\pm$ 0.34 prey species (range: 1–6 species; Fig. 3), with the majority (61%) of the fecal pellets containing only one or two species of insects. The 39 species of insects detected in fecal pellets belonged to nine insect orders (Fig. 4C; Table S3). Thirty-one of these species had a detection frequency of 1 (Fig. S3). We detected one species of cricket (*Gryllus rubens*) in 35% of the fecal samples ($n = 8$ pellets). The remaining species were detected in fewer than five pellets each. Lepidoptera comprised 32.1%, Orthoptera 17.8%, Diptera 21.4%, and Coleoptera 17.9% of the insect species identified in fecal pellets. The remainder belonged to Blattodea, Ephemeroptera, Hemiptera, Neuroptera, and Isopoda. The single fecal sample that contained Isopoda (*Armadillium* sp.) was probably due to environmental contamination since these species are terrestrial and have not been observed on the surface of the turbine tower where they might be exposed to bat predation.

### Comparing diversity between species and sample types

Eastern red bat stomachs had a Simpson's diversity index of 0.898, hoary bats 0.878, and eastern red bat fecal pellets 0.973, indicating that both species have similar diversity of insects in their stomach contents; however, fecal samples had slightly higher diversity than the stomach samples.

As we analyzed more bat stomachs and fecal pellets, we continued to identify insect species that we had not previously found in our study (Fig. 5). Our discovery rate suggests that we would have continued to discover more species of insects with larger sample sizes of bat stomachs and fecal pellets, suggesting that our analysis may only reveal a fraction of the insect species that bats are eating in a night at our study site.

### Comparing stomach and fecal contents to insect surveys

To determine whether insects frequently found in bat stomachs were present at wind turbines, we compared the insect species detected most frequently in the stomach contents to the insect surveys. Because 2012 insect surveys did not incorporate species identification, we only compared our stomach content results to 2013 and 2015 insect surveys. We omitted any insects that were found in <5 stomachs because so many species from our genetic
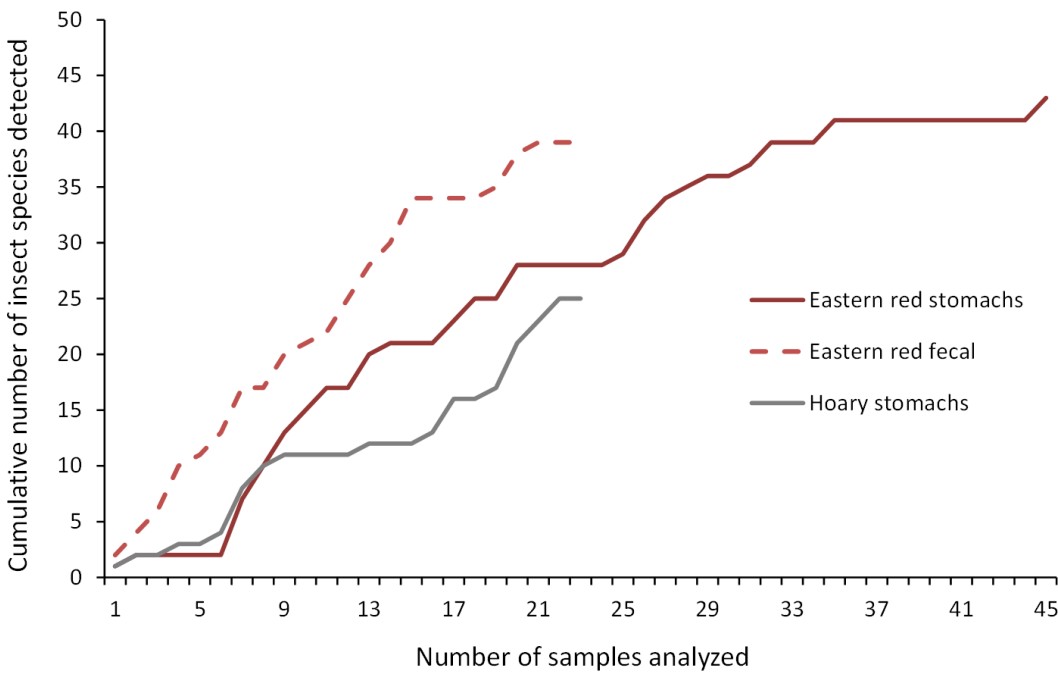

**Figure 5 Discovery rate of new insect species in bat stomachs and fecal pellets.** Discovery rate of new insect species by number of bat stomachs and fecal pellets analyzed from the Wolf Ridge wind farm.

**Table 1 The insect species most commonly found in bat stomach and fecal pellet samples compared to their relative abundance in insect surveys at wind turbines at the Wolf Ridge wind farm.** Crop pest information comes from *Cole & Jackman (1981)*.

| | | | Found in proportion of samples | | | Present at Wolf Ridge | |
|---|---|---|---|---|---|---|---|
| Common name | Scientific name | Crop pest | Eastern red bat stomachs (*n* = 45) | Eastern red bat fecal pellets (*n* = 23) | Hoary bat stomachs (*n* = 23) | Proportion of survey nights (*n* = 29) | Total count |
| Field cricket | *Gryllus spp.* | No | 0.644 | 0.348 | 0.783 | 0.724 | 217 |
| Fall armyworm moth | *Spodoptera frugiperda* | Yes | 0.244 | 0.087 | 0.304 | 0.586 | 128 |
| Necklace veneer moth | *Euchromius ocelleus* | No | 0.089 | 0.000 | 0.130 | 0.207 | 39 |
| Bent-winged owlet moth | *Bleptina caradrinalis* | No | 0.067 | 0.000 | 0.130 | 0.069 | 3 |
| Corn earworm moth | *Helicoverpa zea* | Yes | 0.067 | 0.000 | 0.087 | 0.345 | 38 |

analysis were single-stomach detections. This left us with four species of moths (*E. ocelleus*, *B. caradrinalis*, *H. zea*, and *S. frugiperda*) and two unidentified species of cricket (*Gryllus spp.*). We found these insects in the stomachs of both eastern red bats and hoary bats (Table 1). Of these commonly eaten insect species, we documented most at wind turbines in 2013 and all of them at wind turbines in 2015 (Table 1). Furthermore, all of these species were found at wind turbines on multiple nights throughout the survey period in both 2013 and 2015 (Table 1).

Fecal analysis also had a high number of single-pellet detections, so we only compared insect species detected in ≥5 eastern red bat fecal pellets to the results of our insect surveys.

Crickets (*Gryllus spp.*) were the only species of insect consistently detected in eastern red bat fecal pellets; crickets were also collected on multiple nights throughout the insect survey period in both 2013 and 2015.

## DISCUSSION

Our study provides strong support for the hypothesis that bats are using wind turbines as a foraging resource. We recorded acoustic foraging activity, including feeding buzzes, in the immediate vicinity of wind turbines for all six bat species known to be present at this site (listed as prediction 1 in *Cryan & Barclay, 2009*). We know that light-colored turbines attract aggregations of insects (*Long, Flint & Lepper, 2011*), and found that the orders of insects present at turbines were in relatively consistent proportions from one year to the next (listed as prediction 3 in *Cryan & Barclay, 2009*). Furthermore, for the two bat species most commonly found in fatality searches at this site, we demonstrated that they had full or partially full stomachs, indicating that they were foraging just prior to their deaths (listed as prediction 4 in *Cryan & Barclay, 2009*). We collected bat fecal pellets from turbine structures (e.g., between door slats), which indicates that bats were likely using the turbines as night roosts between successive foraging bouts (*Bennett, Hale & Williams, 2017*). We also demonstrated that the insect species eastern red and hoary bats consistently preyed upon were also present at wind turbines (listed as prediction 5 in *Cryan & Barclay, 2009*).

The presence of foraging or approach phase calls and feeding buzzes from all six species known to be in the study area demonstrates that bats are actively foraging in close proximity to the wind turbines. More specifically, feeding buzzes recorded from all six species at acoustic detectors placed on the nacelle (63 of the 113 feeding buzzes recorded) provides evidence that bats are successfully capturing prey items in the rotor swept zone. This latter observation is important as it indicates that bats are foraging at heights at which they are susceptible to collision with rotating turbine blades. It is also possible that we overestimated foraging activity based on feeding buzzes as these types of acoustic signals are also indicative of bats attempting to locate landing sites (e.g., *Melcón, Denzinger & Schnitzler, 2007*) or drink water (*Griffiths, 2013*). The relative frequency of these activities may warrant additional research as it has been shown that bats will roost on wind turbines (*Bennett, Hale & Williams, 2017*) and it has been hypothesized that bats may misperceive the smooth surfaces of wind turbine tower monopoles to be water (*McAlexander, 2013*).

We found Lepidoptera and Coleoptera in consistently high proportions at turbines at Wolf Ridge in July and August over three years (2012, 2013, and 2015), which suggests that food resources for insectivorous bats were consistently available. Overall, the patterns of abundance in the three survey years remained consistent, despite differences in survey methods. The proportions of Lepidoptera in 2013 and the proportions of Hemiptera in 2015 had much wider confidence intervals than those orders in other years, which could be due to survey methods or differences in other variables that contribute to insect emergence patterns or abundances (e.g., moonlight and weather). While Orthoptera were not as abundant as Lepidoptera or Coleoptera, we consistently caught *Gryllus spp.* each year.

The majority of the bat stomachs in our study were full, or partially full, also indicating that the bats were likely killed while they were foraging. Stomach fullness is a good

indicator of recent foraging, because insectivorous bats typically forage until they have consumed somewhere between one-quarter of their body weight to their full body weight in insects, after which they go to a nearby night roost to digest (*Barclay, Dolan & Dyck, 1991*; *Kunz, Whitaker Jr & Wadanoli, 1995*; *Knight & Jones, 2009*; *Ammerman, Hice & Schmidly, 2012*; *Gonsalves et al., 2013*; *Lison, Palazon & Calvo, 2013*; *Montero & Gillam, 2015*). The fecal pellets included in our study were collected from within structures associated with the turbine towers, providing evidence that eastern red bats were roosting on the turbine structures, likely between successive foraging bouts at night (*Bennett, Hale & Williams, 2017*).

Genetic analysis of dietary habits for insectivorous bats potentially allows for better prey identification, often to species level, compared to the morphological analysis methods used in previous investigations (*Clare et al., 2009*; *Valdez & Cryan, 2013*). For both eastern red and hoary bats, we found between one and six species of insects in their stomachs and fecal pellets, which is consistent with other studies (*Clare et al., 2009*; *Whitaker Jr, McCracken & Siemers, 2009*). If bats consume up to their own body weight in insects per night, the results of our study (and other studies of bat stomach contents and feces) probably represent only a fraction of the bats' nightly diets (*Barclay, Dolan & Dyck, 1991*). This could explain why so many of the insect species we identified were only detected once. We expect that if we had included more bat stomachs and fecal pellets in our analysis, we would have continued to identify additional insect species in the diets of these bats.

Lepidoptera dominated the diets of both eastern red bats and hoary bats, adding to the body of research showing that moths make up a large part of the diet of insectivorous bats (e.g., *Carter et al., 2003*; *Valdez & Cryan, 2009*; *Clare et al., 2009*; *Reimer, Baerwald & Barclay, 2010*; *Zeale et al., 2011*; *Valdez & Cryan, 2013*). Bats digest moths more efficiently than other types of prey, which could explain this abundance in their diets (*Barclay, Dolan & Dyck, 1991*). Despite the differences in the orders found in the stomach contents of eastern red and hoary bats at Wolf Ridge, the two species had similarly high Simpsons' indexes of diversity, indicating that both species eat a wide range of prey. On the other hand, eastern red bat fecal pellets had a higher Simpson's index of diversity than the stomach contents of either species, perhaps because fecal pellets contain a mix of insects from ≥1 foraging bouts.

We found only three orders of insects in the hoary bat stomach contents, which primarily consisted of moths and *Gryllus spp*. These findings are consistent with the results of the hoary bat fecal analysis conducted in Texas by *Valdez & Cryan (2013)*. In that study, there was evidence of Coleoptera in fecal pellets, but Lepidoptera and Orthoptera comprised a larger percentage of the volume of the fecal pellets and had higher detection frequencies overall.

We found nine orders of insects in eastern red bat fecal pellets, primarily consisting of Lepidoptera, Coleoptera, Diptera, and Orthoptera. High percentages of Lepidoptera, Coleoptera, and Orthoptera are consistent with the results of other studies mentioned above. While 10 different species of Diptera were detected, only one was identifiable to the species-level. The presence of a diversity of Dipteran species in the fecal pellets was different than what we observed in the stomach contents. The Dipterans consumed by bats

included several families that feed on plant tissue, including Agrommyzidae (leaf miner flies), Ephydridae (shore flies), and Tephritidae (fruit flies).

Five species of insects met our criteria for consistent detection in bat stomachs (≥5 bat stomachs) and we documented all five in our insect surveys at the turbines at Wolf Ridge, which provides further support for the foraging attraction hypothesis. We found four of the five insect species in both 2013 and 2015, and all species were detected on multiple survey nights in any given year. In contrast, only one species of insect met our criteria for consistent detection in eastern red bat fecal pellets.

We consistently found field crickets, *Gryllus spp.*, in the stomachs of both eastern red and hoary bats as well as in the eastern red fecal pellets, indicating that crickets are an important food source for bats foraging at Wolf Ridge. Ours was not the first study to document *Gryllidae* crickets in bat diets in Texas (*Valdez & Cryan, 2013*), and several explanations have been posited about how and why bats eat crickets. Field crickets have been observed to be attracted to light, and may therefore concentrate at the white turbine towers that are often illuminated by the moon (*Tinkham, 1938*; *Long, Flint & Lepper, 2011*; *Thomson, Vincent & Bertram, 2012*). Additionally, bats may be able to hear crickets chirp, making them easy prey to target. Eastern red and hoary bats are aerial insectivores, meaning they eat on the wing, but studies have suggested that they may glean crickets from surfaces such as canyon walls (*Easterla & Whitaker Jr, 1972*) and turbine towers (*Valdez & Cryan, 2013*). Crickets are primarily terrestrial, but within populations some crickets possess a longer-wing mutation that makes them better flyers; perhaps the crickets found in our diet analysis possess this mutation (*Olvido, Elvington & Mousseau, 2003*; *Valdez & Cryan, 2013*). We do not have direct observations of bats capturing crickets or other insects at turbine towers, but a recent study by *Rydell et al. (2016)* reported that bats killed at wind turbines in southern Sweden had consumed diurnal flies as well as flightless insect taxa, indicating that bats were able to effectively capture prey resting on the turbine surface or from the air near the surface after being disturbed by the bats. These recent diet analyses (this study, *Rydell et al., 2016*) in combination with several published observations of bats making close "investigative" approaches of turbine towers (e.g., *Horn, Arnett & Kunz, 2008*; *McAlexander, 2013*; *Cryan et al., 2014*), suggest that at least some aerial-hawking bat species may be able to capture insects that rest on wind turbine tower surfaces.

We consistently found two species of crop pests in the stomachs of the bats in this study. This result underscores the important pest-management role insectivorous bats play in the ecosystem and in the agriculture industry (e.g., *Boyles et al., 2011*). The most common moth species we found in the bat stomachs, *S. frugiperda*, or the fall armyworm moth, migrates from South Texas and Mexico to North Texas (*Knutson, 2008*; *Westbrook, 2008*). This species is a crop pest, primarily on Bermuda grass (*Cynodon dactylon*), wheat (*Triticum spp.*), and rye grass (*Lolium spp.*), but attacks other crops as well and is most abundant in Texas from August through November (*Knutson, 2008*). In addition to *S. frugiperda*, we also consistently found the corn earworm moth (*H. zea*; *Cole & Jackman, 1981*).

## CONCLUSION

The results of this study and others are providing compelling evidence that at least some bat species are foraging at wind turbines, that foraging ecology in a broad sense is likely contributing to bat fatalities at wind turbines worldwide, and that potential attraction to insect aggregations at wind turbines may interfere with the ability of pre-construction activity surveys to predict risk to bats. For example, if turbines reliably attract insects which in turn attract bats, then pre-construction bat activity surveys at potential wind facilities could drastically underestimate post-construction bat fatality rates. If reliable and abundant foraging opportunities are available for migrating bats at wind turbines and the resulting foraging activity increases risk of barotrauma or collision with rotating blades, then future efforts must focus on technological innovations (e.g., acoustic deterrents) and/or operational changes (e.g., raising the cut-in speed on low wind speed nights) to reduce bat mortality at wind turbines.

While the focus of this study was on the foraging attraction hypothesis, bats may be coming into contact with wind turbines for a variety of reasons that include other sources of attraction in addition to coincidental and random explanations (*Cryan & Barclay, 2009*). Moreover, the attractors likely vary in relative importance by species and over time and geographical space. Thus, this multitude of ultimate causes for bats approaching wind turbines towers, including the aforementioned aggregations of insects, makes devising and implementing effective means to reduce bat fatality without incurring losses in power production a critical element of continued wind power development.

## ACKNOWLEDGEMENTS

We would like to thank NextEra Energy Resources for providing access to and logistical support at Wolf Ridge Wind, LLC. We are grateful to D Cochran and C Bienz, who conducted insect surveys and carcass searches in 2012 and 2013, respectively. Special thanks to C Lindsey, M Paulsen, A Avrin, C Tolle, M McQueen, K Hoenecke, and P Ryan for help with insect surveys in 2015, and to K Hoenecke and M Melton for help dissecting bat stomachs. For assistance with the acoustic monitoring we thank C Sutter, A Costello, A McAlexander, C Lindsey, and R Conley. We also thank the numerous field technicians who participated in the fatality searches at Wolf Ridge over the years.

### Funding

This work was supported by the TCU-NextEra Energy Resources Wind Research Initiative with funding provided to Amanda M. Hale and Victoria J. Bennett (P23186, P23113) and by an Adkins Fellowship from the Biology Department at TCU to Cecily F. Foo. The funders had no role in study design, data collection and analysis, decision to publish, or preparation of the manuscript.

## Grant Disclosures

The following grant information was disclosed by the authors:
TCU-NextEra Energy Resources Wind Research Initiative: P23186, P23113.

## Competing Interests

The authors declare there are no competing interests.

## Author Contributions

- Cecily F. Foo conceived and designed the experiments, performed the experiments, analyzed the data, wrote the paper, prepared figures and/or tables, reviewed drafts of the paper.
- Victoria J. Bennett conceived and designed the experiments, performed the experiments, analyzed the data, contributed reagents/materials/analysis tools, wrote the paper, prepared figures and/or tables, reviewed drafts of the paper.
- Amanda M. Hale conceived and designed the experiments, analyzed the data, contributed reagents/materials/analysis tools, wrote the paper, prepared figures and/or tables, reviewed drafts of the paper.
- Jennifer M. Korstian and Alison J. Schildt performed the experiments, analyzed the data, reviewed drafts of the paper.
- Dean A. Williams conceived and designed the experiments, analyzed the data, contributed reagents/materials/analysis tools, wrote the paper, reviewed drafts of the paper.

## Data Availability

The raw data was supplied as Supplemental Files.

## Supplemental Information

Supplemental information for this article can be found online at http://dx.doi.org/10.7717/peerj.3985#supplemental-information.

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
