# Peer review of "Increasing evidence that bats actively forage at wind turbines"

_PeerJ, doi:10.7717/peerj.3985_

## Round 0.1 · original submission · Major Revisions

I agree with the Referees that this is a nice manuscript that is well-written and will make a solid contribution to the literature. The major comment/concern that I have echoes those made by both referees - the data you have collected suggest that bats forage around turbines. However, I am not convinced by these analyses that the data provide "compelling evidence" to support the "attraction" hypothesis of bat foraging behavior (bats could simply be foraging elsewhere with similar foraging resources). I agree with both Referees that this statement needs to be tempered somewhat, particularly in the Conclusion section. This is my only major concern in the manuscript, although you should carefully review and address the comments made by both Reviewers in your resubmission (be sure to address the comments of Reviewer 1 made to the manuscript text - in particular those having to do with insect ID from DNA data).

Finally, I want to make it clear that the revisions I am requesting to the text are best described as "slightly-more-than-minor" versus "major". While there is no button for me to click for this review outcome, I think that the authors should be able to handle all of these quickly. Below I have some additional minor comments on the tex.

I look forward to your resubmission.

-Brant Faircloth


Minor Comments
* * *
line 71: there is an extra paren. after "two"

line 246: unclear why you picked "at least nine" colonies. just add some extra text describing how you ended up with this starting value.

line 265-268: this follows-on from previous comment, but determining the minimum number of clones needed *after* picking them seems backwards. would be helpful to explain why you took this approach.

line 282: be explicit here about the number of colonies picked (again, was it 9? and, why?)

line 295-296: I agree that finding species in bat stomachs that were also on-site suggests supports the foraging attraction hypothesis, but it is also difficult, with the current study design, to determine whether bats were attracted to forage in a given site. Would temper statement here, somewhat.

line 308: please include show these data (supplement is fine)

line 338: 68 stomachs, but out of how many, total?

line 340-343: what factors affected your ability to sequence clones? please add details.

line 371: again, 23 fecal pellets, out of how many?

line 523: I do not agree that this study provides compelling evidence that bat species are "attracted" to wind turbines. This is the main statement that needs tempering (but also see comments on lines 295-296 as well as other reviews).

Reviewer 1 ·

Basic reporting

No comment

Experimental design

One concern about the experimental design is that methods for species identification for acoustic calls needs to be clarified on if all species were identified using auto-ID functions of the software or if there were comparisons with call libraries during manual identification of call files.

Another concern about the experimental design relates to the data collected from insect trapping being compared to genetic data of fecal and stomach contents. It appears to me that insects from trapping were not identified to the species level using genetic analyses and therefore, those data can't be compared directly to species identification of the genetic material at the same taxonomic level.

See specific comments on the manuscript.

Validity of the findings

It needs to be clarified how fecal samples were collected with details of where guano was being obtained. Were these guano samples in small piles or were there fecal pellets stuck to the turbine/door as if the guano was had fallen from a bat in flight. Isopods discovered in the fecal samples, as found in the supplemental tables, raises the question on the validity of identified insects consumed by bats. This is particularly so given that these arthropods are usually on the ground or under substrate and generally photophobic and unlikley to be attracted to the turbines themselves. In light of this, there are possible explanations for its presence in your samples and the need for caveats.

See specific comments on the manuscript.

Additional comments

This is a great paper that uses a variety of methods to detect the presence of foraging behavior by bats at wind energy facilities! However, I feel that there are some missed opportunities when you don't present more on the acoustic data for the other bat species that show a greater amount of feeding activity than hoary bats and red bats.

I do like the fact that there is a comparison of insects captured at the sight vs. the stomach and fecal samples of bats. However, the resolution in species identification from the genetic analyses is difficult to compare to the trapped insects when the same methods aren't employed.

I think some of the conclusions related to your findings are fair to speculate but may not be the entire picture of what is happening in this system. I did play devil's advocate by pointing out other options or reasons for explanation that could also be possible. These are addressed directly in the manuscript.

Annotated reviews are not available for download in order to protect the identity of reviewers who chose to remain anonymous.

Reviewer 2 ·

Basic reporting

no comment

Experimental design

no comment

Validity of the findings

The statement in the discussion “Our study provides strong support for the hypothesis that bats are using wind turbines as a foraging resource” can be interpreted in two ways: 1) bats forage around turbines; 2) turbines affect bat behavior and/or resources such that the area is preferred or used more than other areas without turbines.

The correspondence between insect composition in light traps and those in bat stomachs and pellets is very suggestive that bats forage in and around wind turbines. It is also certainly possible that turbines can attract insects which in turn attract bats. Alternatively, the correspondence may simply reflect a situation where bats are foraging on available insects (both near and far from turbines) and the turbines do not affect this relationship in any meaningful way. That is, while the study does show that interpretation 1 is valid, it is harder for this reviewer to feel that a strong case has been made for interpretation 2. Foraging behavior at turbines has been observed both visually and acoustically in other studies, and Foo et al. contribute additional evidence that this activity is also prevalent in their study area. (The statement that eastern red bats were likely roosting on the turbine structures after foraging nearby is a particularly noteworthy observation.) However, the study was not designed to falsify the hypothesis that turbines serve as a focal resource (although they may well be). To accomplish this, it would be necessary to sample both bat activity and insect prevalence at sites distant from turbines, and perhaps also assess fecal pellets obtained from bats captured at these locations with those collected at turbines. From this a comparison of the proportion of bat calls with approach and terminal phase components recorded at locations distant could be made with those sampled close to turbines. If proportions were similar, it would indicate that turbines are not necessarily attracting bats and they simply encounter these features as they are foraging across a landscape; that is, foraging activity is similar. A study that applied this method to acoustic samples is Jameson and Willis (2014) who observed “a smaller proportion of feeding buzzes at [meteorological] towers compared to woodlots during all study periods despite the dramatic increase in overall activity at towers during autumn” and concluded that bat presence was likely associated with visual attraction to tall structures for purposes of social interactions.

I applaud authors for a solid study showing that there is, as stated in the manuscript title “increasing evidence that bats actively forage at wind turbines”. However, the authors may wish to consider tempering statements such as “abundant foraging opportunities continue to attract migrating bats to wind turbines” with a caveat that “attraction”, that is, relatively greater foraging use of turbine areas was not experimentally determined nor specifically the focus of the study.

Additional comments

It would be useful to see the night-to-night correspondence in stomach samples relative to insect trapping. High correspondence would indicate that utilization matched availability (at least for certain orders), and provide strong evidence of prey capture at or near the wind site. Low correspondence would indicate prey capture at a distance from a site, or perhaps a low within-night match of utilization-availability. These results could be added to the supplementary materials.

---

## Round 0.2 · accepted · Accept

Nice job on the revision. I am accepting this manuscript, but please, while in Production:

1. Please alter use of "indicate" to "suggest" (or similar) in Lines 37.

2. I think I misread this the first time around (sorry!). Now, there is a missing parenthesis after "of the two" on line 98. Basically, this a little hard to read because you have nested parentheses - perhaps brackets would be better to offset the main parenthetical.

3. I would use "suggest" in place of "indicate" in line 117.

Reviewer 1 ·

Basic reporting

N/A

Experimental design

N/A

Validity of the findings

N/A

Reviewer 2 ·

Basic reporting

No comment.

Experimental design

No comment.

Validity of the findings

The primary concern this reviewer had with the initial ms was that the authors were making too strong a claim that bats are attracted to turbines for foraging. In the revised ms and response the authors acknowledge this and have made changes to the text that temper this claim while still indicating that it remains a phenomenon that needs further investigation. The authors did demonstrate that foraging does occur at turbines, and correctly conclude that foraging opportunities may be one reason that bats are active at those locations.